

**Estimation and Evaluation of COSMIC Radio Occultation Excess Phase Using**
**Non-differenced Measurements**
Pengfei Xia*, Shirong Ye*✉, Kecai Jiang, Dezhong, Chen
*GNSS Research Centre, Wuhan University, Wu han, 430079,China*
*Correspondence to*: Ye Shirong (srye@whu.edu.cn)
**Abstract**. In the GPS radio occultation technique, the atmospheric excess phase (AEP) can be
used to derive the refractivity which is an important quantity in numerical weather prediction. The
AEP is conventionally estimated based on GPS double-differenced or single-differenced
techniques. These two techniques, however, require the reference link data in the data processing
increasing the complexity of computation. In this study, a non-differenced (ND) processing
strategy is proposed to estimate the AEP. To begin with, we used PANDA (Positioning and
Navigation Data Analyst) software to perform the precise orbit determination (POD) for the
COSIMC (The Constellation Observing System for Meteorology, Ionosphere and Climate)
satellite to acquire the position and velocity of the center of mass of the satellite and the
corresponded receive clock offset. The bending angles, refractivity and dry temperature profiles
are derived from the estimated AEP by the ROPP (Radio Occultation Processing Package)
software. The ND method is validated by the COSMIC products in typical rising and setting
occultation events. Comparison results indicate that RMS (root mean square) errors of relative
refractivity differences between ND-derived and "atmPrf" profiles are better than 4% and 3% in
rising and setting occultation events, respectively. In addition, we also compared the relative
refractivity bias between ND-derived and "atmPrf" profiles of globally distributed 200 COSMIC
occultation events on December 12, 2013. The statistic results show that the average RMS relative
refractivity deviation between ND-derived and COSMIC profile is better than 2% in the rising
occultation event, and it is better than 1.7% in setting occultation event. Moreover, the observed
COSMIC refractivity profiles from ND processing strategy are further validated using European
Centre for Medium-Range Weather Forecasts (ECMWF) analyses data, and the results indicate
that non-differencing reduces the noise level on the excess phase paths in the lower troposphere
compared to single difference processing strategy.
**Keywords:** Radio Occultation; Excess Phase; Precise Orbit Determination; GPS
**1. Introduction**



The radio occultation (RO) technique was first applied in the field of astronomy for detecting the
state of the planet's atmosphere (Kursinski et al., 1997). With the development of GPS
meteorology, the space-based GPS radio occultation is regarded as a valuable data source for
atmospheric change studies (Rocken et al., 1997; Kursinski et al., 1997; Hajj et al., 2002; Beyerle
et al., 2005). Since the GPS/MET (Global Positioning system/Meteorology) mission conducted a
number of successful measurement experiments from 1995 to 1997 (Ware et al., 1996; Rocken et
al., 1997), some low Earth orbiting (LEO) satellites, such as: CHAMP, GRACE, COSMIC and
MetOp-A (Wickert et al., 2001, 2005; Rocken et al., 2000; Wilson et al., 2010), have begun to
equip RO instruments facilitating the development of RO technique. In the GPS occultation
technique, the atmospheric refractivity is an important quantity in the numerical weather
prediction (Esteban et al., 2013). The atmospheric excess phase (AEP) can be used to derive
the bending angles of the GPS rays and further to obtain the refractivity from the bending angles.
Thus, the retrieval accuracy of refractivity is quite dependent on the quality of the estimated AEP.
Conventionally, the AEP is determined using two kinds of differential technique, i.e. double-
difference (DD) (Sokolovskiy et al., 1996; Kursinski et al., 1997; Rocken et al., 1997; Hajj et al.
2002) and single-difference (SD) (Wickert et al., 2002; Schreiner et al., 2005), in which various
errors can be eliminated from the differencing of GPS observations.  The double-differenced
method requires additional data from the ground receiver and the reference GPS satellite to
remove the oscillator errors of the transmitter and the LEO receiver. This processing will bring the
error sources to AEP from the ground data and the reference GPS satellite, such as multipath error,
residual ionospheric and troposphere noise, thermal noise, and so on (Schreiner et al., 2010).
Single-difference processing has a potential advantage over double-differencing since it can
eliminate the ground observation data error (Schreiner et al., 2010). However, single-difference
excess phases also suffer the noise sources from the reference link data. Compared with the two
differential techniques, non-differenced (ND) method does not require the reference link data,
which reduces the complexity in data processing. Besides, ND processing can potentially obtain
lower noise AEP by utilizing prior estimated LEO and GPS clocks (Beyerle et al., 2005; Schreiner
et al., 2011). Beyerle et al (2005) firstly proposed the idea of ND technique to estimate the AEP,
and  successfully  analyzed  the  GRACE-B  satellite  to  correct  the  effect  of  receiver  clock  by





interpolating the temporal resolution of 30s GRACE-B's receiver clock solutions into 20 ms. Their
results show that there is a good agreement of the refractivity between ND-derived and SD-
derived in the upper troposphere and lower stratosphere. More significantly, ND technique can
reduce the noise level and yield less-biased refractivity in the lower troposphere compared to SD
technique. However, ND technique needs a LEO receiver with an ultra-stable oscillator (Beyerle
et al. 2005; Schreiner et al., 2010). Therefore, "single difference" method is still widely utilized to
get AEP in the current international various GPS RO data processing centers (Bi et al., 2012).
The accuracy of ND-derived refractivity mainly depends on the quality of clock error.
COSMIC (The Constellation Observing System for Meteorology, Ionosphere and Climate)
satellites' orbits and clock solutions are provided at a temporal resolution of 30s. Subject to the
effect of COSMIC receiver oscillator, a lot of noises will be introduced when interpolating 30s
clock solutions into 20 ms. In this study, we adopt PANDA (Positioning And Navigation Data
Analyst) software to determine the COSMIC satellite orbit and obtain the receiver clock error at
an interval of 1s. Then, the AEP is extracted utilizing the ND technique. Additionally, the
refractivity and dry temperature will be derived from the AEP based on ROPP (Radio Occultation
Processing Package) software. Finally, we compared the ND-derived refractive with the "atmPrf"
profiles provided by UCAR/CDAAC (University Corporation for Atmospheric
Research/COSMIC Data Analysis and Archive Center). Moreover, the ND-derived refractivity
profiles were further evaluated by comparing with the analyze field data of ECMWF (European
Centre for Medium-Range Weather Forecasts).
The rest of the paper is organized as follows. Section 2 introduces the principle of estimating
AEP using non-difference method. Section3 describes the processing of LEO precise orbit
determination using PANDA software. Section 4 presents the validation of ND-derived
refractivity. The conclusions are included in Section5.
**2    Non-Differencing Method**
The signals of the GPS occulting satellite are recorded by the RO receiver aboard the spacecraft in
50 Hz during the occultation event. The carrier phase measurements with repaired cycle slip can
be expressed as (Schreiner et al., 2010):





$$Li_r^s(t_r) = c \cdot \delta t_r(t_r) + c \cdot \delta t_{r,rel}(t_r) + \rho_r^s(t_r) + \rho_{r,rel}^s(t_r) + c \cdot \delta t^s(t_r - \tau_r^s) +$$
$$c \cdot \delta t_{rel}(t_r - \tau_r^s) + \delta p_{r,ion}^s(t_r) + \delta p_{r,trop}^s(t_r) + \lambda_i \cdot N_{amb} + V_{pco} + \varepsilon \tag{1}$$

where $i=1,2$; $t_r$ and $\delta t_r$ indicate the receive time and the deviation between receiver time and right

time at receive time, respectively; $c$ is the speed of light in vacuum; $\delta t_{r,rel}$ represents the offset

between right time and coordinate time at the receiver owing to special and general relativity; $t^s$

and $\delta t^s$ denote the transmitted time and the deviation between proper time and satellite time at

transmit time, respectively; $\delta t^s_{rel}$ expresses the offset between right time and coordinate time at

satellite; $\rho^s_r$ is the geometric range between GPS satellite and COSMIC satellite; $\delta\rho^s_{r,rel}$ means

gravitational delay correction; $\delta\rho^s_{r,ion}$ and $\delta\rho^s_{r,trop}$ signify the ionospheric delay correction and

tropospheric delay correction, respectively; $\tau^s_t$ indicates the light travel time in vacuum; $N_{amb}$

represent phase ambiguity; $V_{pco}$ means the antenna phase center offsets; $\varepsilon$ is phase noise.

The above equation neglects multipath errors, carrier phase wind-up and so on. The position and

the clock error of GPS satellite are provided by the International GNSS Service (IGS). In addition,

the $\tau^s_t, \delta t^s_{rel}$ and $\delta\rho^s_{r,rel}$ can be modeled utilizing the following equations (Schreiner et al., 2010):

$$\tau_r^s = \frac{\left(\rho_r^s(t_r) + \delta p_{r,rel}^s\right)}{c} \tag{2}$$

$$\delta t_{rel}^s = -2\frac{r^s \cdot v^s}{c^2} \tag{3}$$

$$\delta p_{r,rel}^s = \frac{2GM_E}{c^2} \ln\left(\frac{r^S + r_r + \rho_r^s}{r^S + r_r - \rho_r^s}\right) \tag{4}$$

where $r^s$ and $r_r$ mean the position and velocity vectors of the GPS satellite at signal transmit time

in an earth-centered inertial (ECI) reference frame (Ashby 2003); $G$ denotes Newton's

gravitational constant; $M_E$ expresses the earth's mass; $r^s$ and $r_r$ represent the GPS satellite and

receiver radial positions at the GPS signal transmit and receive times.

The $L1$ and $L2$ channel phase can be combined with satellite position and velocity data to

determine the AEP. Neglecting the influences of ambiguity and time-independent error terms, the

use of ND method to calculate the AEP ($\Delta L_i$) can be modeled as follows (Schreiner et al., 2010):

$$\Delta L_i = \delta p_{r,ion}^s(t_r) + \delta p_{r,trop}^s(t_r) = Li_r^s(t_r) - c \cdot \delta t_r(t_r) - c \cdot \delta t_{r,rel}(t_r) - \rho_r^s(t_r)$$
$$- \rho_{r,rel}^s(t_r) - c \cdot \delta t^s(t_r - \tau_r^s) - c \cdot \delta t_{rel}(t_r - \tau_r^s) - V_{pco} \tag{5}$$

The input $L1$ and $L2$ phase measures of COSMIC RO are provided by the "opnGps" profiles



which can be downloaded in UCAR/CDAAC at a temporal resolution of 20 ms. Besides,
UCAR/CDAAC also supply the COSMIC receiver clock offset "leoClk" profile and the GPS
satellite clock offset "comClk" profile at a temporal resolution of 30 s. Each of the COSMIC
satellites is equipped with the BlackJack GPS receiver, a tiny ionospheric photometer (TIP) and a
tri-band beacon (TBB) (Wu et al. 2005; Schreiner 2005; Montenbruck et al. 2006), and the
Integrated GPS Occultation Receiver (IGOR) is designed by the Jet Propulsion Laboratory (JPL)
and manufactured by Broad Reach Engineering (Schreiner et al. 2011). By analyzing the results of
"leoClk" profiles, we conclude that the turbulence of IGOR receiver clock is relatively serious.
Therefore, estimated AEP at the required temporal resolution of 20 ms could not be interpolated
successfully from the 30 s clock solutions utilizing Eqs.(5).  It will be an effective technique to
deal with this problem by reprocessing the COSMIC satellite orbit and obtaining the high accurate
and high temporal resolution of IGOR clock offset.
**3 COSMIC spacecraft precise orbit determination**
3.1 COSMIC POD processing
The joint Taiwan/US mission COSMIC, including six micro-satellites, was launched on April 17,
2006. Each of the satellites is equipped with a GPS receiver which is installed with four antennas
on the front and back faces of the satellite main frame. Two single-patch antennas, mounted on the
upper part of the main body, are for POD. The other two antennas, dedicated to atmospheric
occultation research, are mounted on the lower part (Hwang et al., 2009). The POD of COSMIC
satellite is an important premise in atmospheric occultation research. At present, UCAR
(University Corporation for Atmospheric Research) provides three kinds of COSMIC orbit
products, i.e. reprocessing products, post-products and real-time products. UCAR/CDAAC
reprocess products adopt Bernese5.2 software as processing tool, and orbit determination method
is improved as well as the processing method of the phase values is more elaborate. The three-
dimensional position average RMS value of overlapping orbit precision is superior to 15 cm, and
the three-dimensional velocity RMS value is better than 0.15 mm/s (CDAAC, 2013). In addition,
Hwang et al., (2009) calculated the COSMIC satellite orbits using Bernese 5.0 software and
differences between their orbit products and those of UCAR are at the level of 10 cm on three axis.



PANDA is satellite positioning and orbit determination software which is an independent
research and development product by satellite navigation and positioning technology research
center of Wuhan University. The software has the ability of processing many kinds of observation
data, such as GNSS (including GPS, GLONASS, GALILEO, and BDS), SLR, KBR, satellite
attitude and so on (Liu et al., 2004). In the paper, the PANDA software is exploited to perform
POD for the COSMIC satellite. The process of this inputs include the COSMIC L1 and L2
pseudo-range and carrier phase data form the HAICH-FARR antenna, COD Final GPS orbits, 5-s
COD-provided transmitter clock offsets from GPS time, LEO attitude information from CDAAC,
earth orientation information and L1and L2 antenna phase center variations. Ionosphere-free phase
observations are utilized based on a post-processing generalized least squares approach to
determine the position and velocity of the center of mass of the LEO satellite as a function of
coordinate time in an ECI reference frame and clock of the LEO every 1 second. In the process,
the POD is calculated over 30-hour data arcs utilizing 1Hz carrier phase observations from
HAICH-FARR antenna. The state vector computed in this processing also take the influence of
dynamic model or estimation method into account, such as the gravitational field, the earth and
ocean tides, tidal dynamic model, solar radiation pressure, experience force and so on.
Furthermore, gravity field uses EIGEN2 model whose order is set to 140, and solar radiation
pressure uses box-wing model.
3.2 POD Precision Evaluation
In the processing of POD, firstly we can obtain the initial COSMIC satellite orbit, state parameters
and mechanical parameters through the pseudo-range single point positioning, then such initial
solutions can be used for orbit integration in order to achieve the purpose of further elaboration.
The next stage of processing is to detect gross errors and cycle slips of the GPS carrier phase data,
then the COMSIC orbit precision will be improved according to the residual errors to iterative
least squares estimate and residual edit operation. Finally, the orbit and clock error will be output
every second. In order to test the accuracy of COSMIC orbits from PANDA software, we selected
53-day satellite-borne GPS observation data from day of year (DOY) 313 to 365 of 2011which is
provided by UCAR/CDAAC. Due to the missing of 3[th] COSMIC satellite data, we only processed



the remaining 5 satellites' data and removed those data with observation time less than 10 hours.
Since the COSMIC satellites are not equipped with laser corner reflector, the accuracy of orbit
determination cannot be evaluated by SLR precision ranging information. We thus mainly
compared our orbit results with UCAR/CDAAC to analyze the precision of orbit determination.
The statistical results of five satellites orbit between PANDA-derived and UCAR/CDAAC-
derived are given in Table 1. In addition, there is a 6-hour overlapping orbit between the tracks
because of the usage of 30 hours data arcs as the orbit determination length. Table 2 presents the
RMS of COSMIC satellites which are compared with overlapping orbits on three axis direction.
**Table 1.** The RMS of COSMIC satellites compared with UCAR/CDAAC on three axis direction

|  | Radial RMS | | Along-track RMS | | Cross-track RMS | | 3-D Root Sum Square | |
| --- | --- | --- | --- | --- | --- | --- | --- | --- |
|  | Pos (cm) | Vel (mm/s) | Pos (cm) | Vel (mm/s) | Pos (cm) | Vel (mm/s) | Pos (cm) | Vel (mm/s) |
| FM1 | 10.82 | 0.117 | 13.29 | 0.115 | 13.91 | 0.134 | 22.07 | 0.212 |
| FM2 | 10.40 | 0.162 | 16.99 | 0.105 | 9.65 | 0.097 | 22.13 | 0.216 |
| FM4 | 7.92 | 0.103 | 12.38 | 0.085 | 11.70 | 0.120 | 18.78 | 0.180 |
| FM5 | 8.89 | 0.123 | 14.39 | 0.097 | 12.48 | 0.087 | 21.01 | 0.179 |
| FM6 | 10.06 | 0.139 | 16.28 | 0.109 | 10.63 | 0.094 | 21.89 | 0.200 |

**Table 2.** The RMS of COSMIC satellites compared with overlapping orbits on three axis direction

|  | Radial RMS | | Along-track RMS | | Cross-track RMS | | 3-D Root Sum Square | |
| --- | --- | --- | --- | --- | --- | --- | --- | --- |
|  | Pos (cm) | Vel (mm/s) | Pos (cm) | Vel (mm/s) | Pos (cm) | Vel (mm/s) | Pos (cm) | Vel (mm/s) |
| FM1 | 5.37 | 0.051 | 6.57 | 0.058 | 6.67 | 0.048 | 10.80 | 0.091 |
| FM2 | 5.30 | 0.058 | 6.55 | 0.054 | 5.63 | 0.049 | 10.13 | 0.093 |
| FM4 | 6.11 | 0.079 | 8.09 | 0.058 | 6.30 | 0.056 | 11.93 | 0.113 |
| FM5 | 4.94 | 0.069 | 8.07 | 0.053 | 7.02 | 0.057 | 11.78 | 0.105 |
| FM6 | 6.91 | 0.079 | 8.58 | 0.077 | 7.35 | 0.057 | 13.25 | 0.124 |

As shown in Table 1, the average 3-D RMS orbit coordinates and velocity differences between
PANDA-derived and UCAR/CDAAC-derived for the 53-day period are 11.98 cm and 0.11
mm · s[-1], respectively. In addition, Table 1 presents that the accuracy of 4[th] COSMIC satellite's
POD is better than the other four satellites. Table 2 shows the average 3-D RMS POD's
coordinates and velocity are better than 6.63 cm and 0.06 mm·s[-1] compared with overlapping
orbits and velocity, respectively. These results suggest that the quality of COSMIC POD generated
from PANDA software is feasibly.
**4 Results validation and analysis**





The UCAR COSMIC Data Analysis and Archive Center data processing center adopted the
method of single-difference to process data of COSMIC radio occultation, meanwhile deposited
the processed excess phase delay into "atmPhs" profile. Atmospheric profiles of bending angle,
refractivity and dry temperature generated from "atmPhs" files were written into "atmPrf"
profiles. These files are freely available for public access http://cdaac-www.cosmic.ucar.edu/. In
order to evaluate the precision of AEP estimated by ND method, the AEPs are obtained using
Eqs.(5) in this study. Then, the ROPP software is implemented to process excess phase data and
deriving profiles of bending angle, dry temperature and refractivity. Finally, the obtained
COSMIC refractivity profiles are compared with "atmPrf" profiles provided by UCAR/CDAAC.
Furthermore, duo to UCAR/CDAAC also offers the moisture profiles generated from ECMWF
analysis and the ERA (Each re-analysis) interim model which collocated with occultation profiles.
So the comparison between ND-derived refractivity and meteorological analysis results are
performed to further validate the results obtained from ND method.
4.1 The results of typical GPS occultation event
Non-differenced processing strategy is utilized to obtain L1 and L2 excess atmospheric phases as
functions of GPS time in an ECI TOD (true equator and equinox of data) reference frame. Inputs
to this processing are: 50Hz L1 and L2 phase measures for the occulting GPS satellite, LEO and
GPS positions, velocities and clock offsets, and antenna phase center information. Then, the AEPs
are calculated using Eqs.(5), randomly selecting two GPS occultation events on December 12,
2013. Table 3 gives the detailed status parameters of the two GPS occultation events.
**Table 3.** Detailed status parameters of the selected two GPS occultation events on December 12, 2013

| Parameter | OCCsat-1 | OCCsat-2 |
|---|---|---|
| GPS | PRN-08 | PRN-28 |
| LEO | CO06 | CO05 |
| start time | 15:07 | 14:12 |
| last time | 123s | 146s |
| status | setting | rising |
| longitude | 3.78˚E | -109.84˚W |
| latitude | 34.06˚N | 26.08˚N |
| quality mark | bad=0 | bad=0 |

As the method introduced by section 2, we first process the two GPS occultation events to
obtain AEPs by ND technique. Afterwards, these atmospheric excess phases and collocated with
occultation "atmPhs" profiles will be used for generating refractivity based on ROPP software,





respectively. We named them as Ref_ND and Ref_Phs, respectively. Using the "atmPrf" profiles
of refractivity provided by UACR/CDAAC as references, ROPP software is validated by
comparing Ref_Phs with atmPrf products. Moreover, the Ref_ND is evaluated by comparing with
Ref_Phs and atmPrf products, respectively.  These results are depicted by Figure 1 and Figure 2.

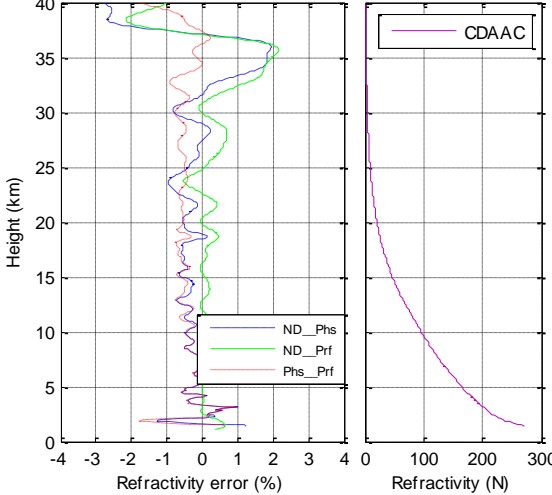

**Fig. 1.** The results of ND_Phs, ND_Prf and Phs_Prf in the setting radio occultation. ND_Phs expresses the relative
refractivity offset between Ref_ND and Ref_Phs; ND_Prf denotes the relative refractivity offset between Ref_ND
and "atmPrf"; Phs_Prf indicates the relative refractivity offset between Ref_Phs and "atmPrf" product; Ref_ND
means the refractivity obtained from ND-derived AEP based on ROPP software; Ref_Phs means the refractivity
obtained from atmPhs profiles based on ROPP software.

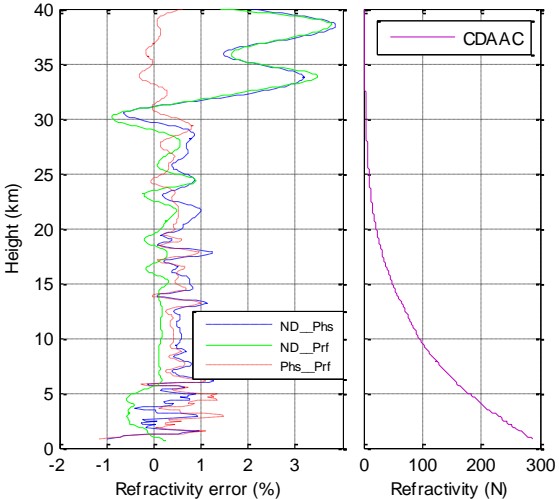

**Fig. 2.** The results of ND_Phs, ND_Prf and Phs_Prf in the rising radio occultation. ND_Phs expresses the relative



refractivity offset between Ref_ND and Ref_Phs; ND_Prf denotes the relative refractivity offset between Ref_ND
and "atmPrf"; Phs_Prf indicates the relative refractivity offset between Ref_Phs and "atmPrf" product; Ref_ND
means the refractivity obtained from ND-derived AEP based on ROPP software; Ref_Phs means the refractivity
obtained from atmPhs profiles based on ROPP software.

Figure 1 and Figure 2 show the results for the setting and rising occultation, respectively. The
red dashed lines are less than ±1.7% under the height of 40m, which verifies the feasibility of
ROPP software. In addition, the green lines are closer to 0 than red lines and blue lines under the
height of 30km while the green line and blue line are gradually increases from the height of 30km
to 40km. Besides, Figure 1 and Figure2 also depict that the refractivity is exponential change with
height, and the refractivity is less than 5N from the height of 30km to 40km.
4.2 The statistic and verified of ND method
The COSMIC RO provides about 1800 RO events per day and the science mission is mainly for
weather, climate, space weather, geodetic research and so on study purpose (Yen et al., 2007; Kuo
et al., 2007). To verify the ND method, we randomly selected 200 RO events on December 12,
2013 to obtain the AEP utilizing Eqs.(5), then deriving profiles of refractivity and dry temperature
through ROPP software which are named R_N and T_N, respectively. Figure 3 shows the
distribution of the 1605 RO events on December 12, 2013 and the selected 200 RO events (blue
triangle).

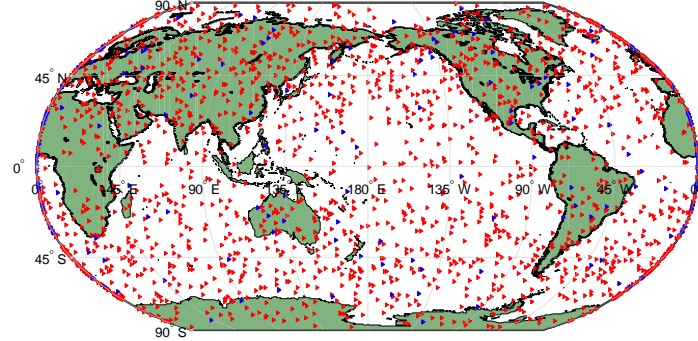

**Fig.3.** The global distribution of COSMIC RO events on December 12, 2013. Blue triangle represents the selected
200 RO events.




There are 112 setting occultation events and 88 rising occultation events in the selected 200
occultation events. ROPP software is implemented to process the "atmPhs" profiles which are
collocated with the selected 200 RO events and derive profiles of refractivity and dry temperature,
which are denoted as R_phs and T_phs, respectively. We then respectively analyzed the setting
occultation events and rising occultation events to obtain the average relative deviation of
refractivity between R_N and R_phs, R_N and atmPrf, R_phs and "atmPrf".

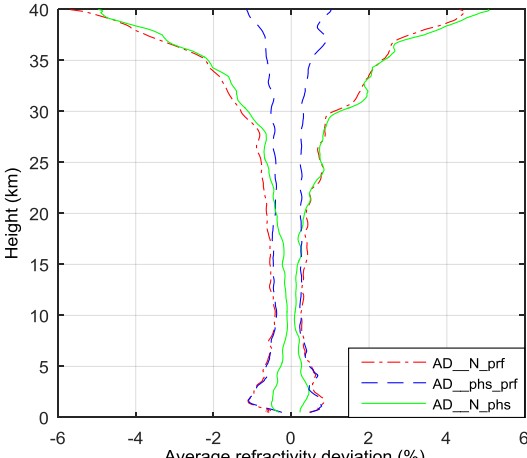


**Fig.4.** The statistical average relative deviation results of the refractivity for setting occultation events.
"AD_N_prf" denotes the average relatively deviation of refractivity between R_N and "atmPrf"; "AD_phs_prf"
expresses the average relatively deviation of refractivity between R_phs and "atmPrf"; "AD_N_phs" represents
average relatively deviation of refractivity between R_N and R_phs; "R_N" means the refractivity derived from
ND-derived AEP for the selected 200 RO events based on ROPP software; Ref_Phs means the refractivity
obtained from atmPhs profiles for the selected 200 RO events based on ROPP software.



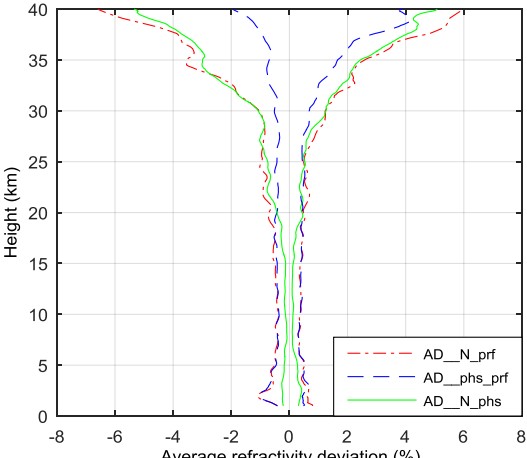

**Fig.5.** The statistical average relative deviation results of the refractivity for rising occultation events.
"AD_N_prf" denotes the average relatively deviation of refractivity between R_N and "atmPrf"; "AD_phs_prf"
expresses the average relatively deviation of refractivity between R_phs and "atmPrf"; "AD_N_phs" represents
average relatively deviation of refractivity between R_N and R_phs; "R_N" means the refractivity derived from
ND-derived AEP for the selected 200 RO events based on ROPP software; Ref_Phs means the refractivity
obtained from atmPhs profiles for the selected 200 RO events based on ROPP software.

As it shows in Figure4 and Figure5, the blue dashed lines are all less than ±1.5% in setting
occultation events and ±4.2% in rising occultation events, which once again verified the feasibility
of ROPP software. In addition, Figure 4 and Figure 5 also indicate that the green lines are closer to
0 than the blue lines and red lines under the height of 20km, while the blue lines are closer to 0
than the red lines and green lines from the height of 20km to 40km. Besides, it is also obviously
that the red lines and the green lines are gradually increases with height from 20km to 40km. At
the same time, the statistics of the refractivity and dry temperature difference between ROPP-
derived and "atmPrf" profiles for the selected 200 RO events are listed in Table 4.

**Table 4.** Summary of the comparison between ROPP-derived and "atmPrf". "ND-phs" repents the comparison
between ND-derived and atmPhs-derived; "ND-prf" repents the comparison between ND-derived and "atmPrf";
"phs-prf" repents the comparison between atmPhs-derived and "atmPrf". (%)

| Parameter | status | ND-phs | ND-prf | phs-prf |
|---|---|---|---|---|
| Refractivity | rising | 1.64 | 1.91 | 0.93 |
| | setting | 1.52 | 1.63 | 0.51 |
| Dry temperature | rising | 2.49 | 3.21 | 1.65 |
| | setting | 2.35 | 2.42 | 0.72 |




289 Table 4 provides the RMS of the comparison between ROPP-derived and "atmPrf" profiles in

290 the rising and setting RO events, respectively. The statistical results indicate that the accuracy of

291 setting RO events is better than the rising RO events. Besides, the RMS of average refractivity

292 differences between ND-derived and atmPrf-derived is better than 2%, and the RMS of average

293 dry temperature deviations between ND-derived and atmPrf -derived is better than 3.3%.

294 4.3 Comparison with ECMWF

295 CDAAC/UCAR center provides the ECMWF analyses produce, including "ecmPrf", "ecmPrf"

296 and "eraPrf" profiles which collocated with radio occultation profiles. Among them, "ecmPrf"

297 profiles contain temperature, pressure and moisture profiles generated from ECMWF analysis

298 with 21 layers; "echPrf" files contain temperature, pressure and moisture profile from ECMWF

299 high precision analysis field data with 88 layers; and "eraPrf" profiles include temperature,

300 pressure and moisture profiles generated from the ERA interim model with 37 layers. Then, the

301 observed COSMIC refractivity profiles utilizing ND method are compared with these three kinds

302 of products, respectively. The mean relative refractivity deviation compared ND-derived and

303 "atmPrf" profiles with ECMWF analyses are shown in Figure 6 with setting RO events and in

304 Figure 7 with rising RO events.





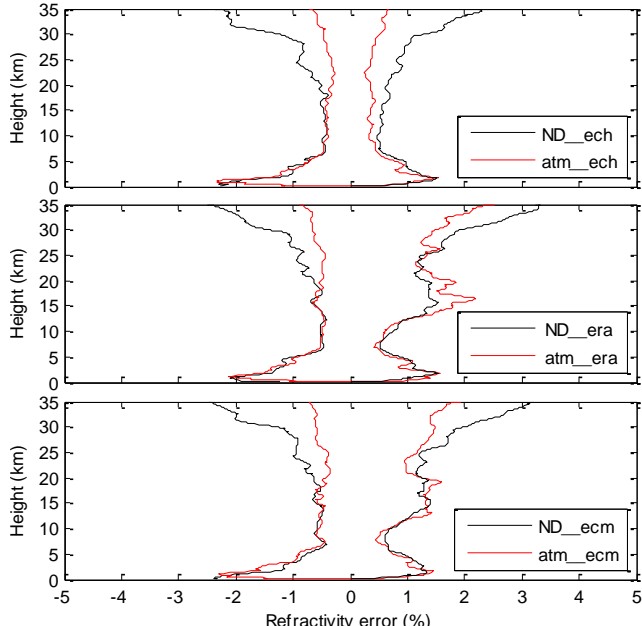

305

**Fig.6.** The mean relative refractivity deviation compared R_N and "atmPrf" with ECMWF analyses in setting RO

events. "ND__ech" represents the average relative refractivity deviation between ND-derived and "echPrf";

"atm__ech" denotes the average relative refractivity deviation between "atmPrf" and "echPrf"; "ND__era" shows

the average relative refractivity deviation between ND-derived and "eraPrf"; "atm__era" expresses the average

relative refractivity deviation between "atmPrf" and "eraPrf"; "ND__ecm" means the relative refractivity deviation

between ND-derived and "ecmPrf"; "atm__ecm" signifies the average relative refractivity deviation between

"atmPrf" and "ecmPrf"; "R_N" means the relative refractivity derived from ND-derived AEP for the selected 200

RO events based on ROPP software.

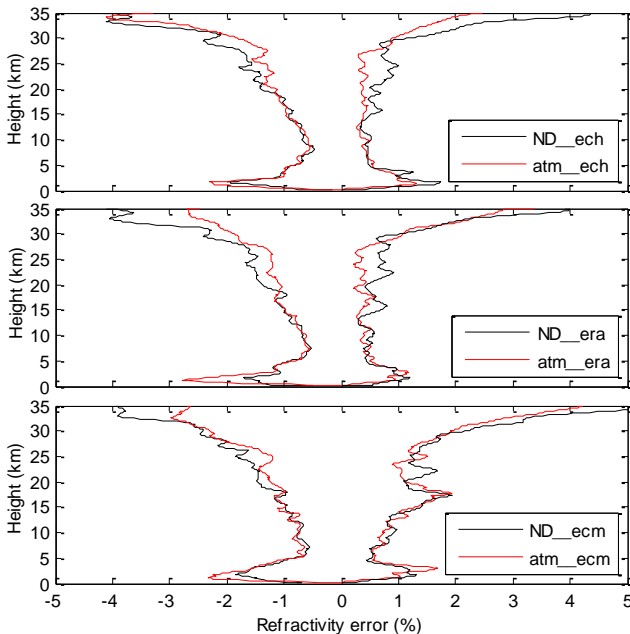

**314**
**315** **Fig.7.** The mean relative refractivity deviation compared R_N and "atmPrf" with ECMWF analyses in rising RO

**316**     events. "ND__ech" represents the average relative refractivity deviation between ND-derived and "echPrf";

**317**   "atm__ech" denotes the average relative refractivity deviation between "atmPrf" and "echPrf"; "ND__era" shows

**318**     the average relative refractivity deviation between ND-derived and "eraPrf"; "atm__era" expresses the average

**319**   relative refractivity deviation between "atmPrf" and "eraPrf"; "ND__ecm" means the average relative refractivity

**320**     deviation between ND-derived and "ecmPrf"; "atm__ecm" signifies the average relative refractivity deviation

**321**   between "atmPrf" and "ecmPrf"; "R_N" means the refractivity derived from ND-derived AEP for the selected 200

**322**                 RO events based on ROPP software.

**323**
**324**     From Figure 6 and 7, it can be seen that the black lines are closer to 0 than the red lines below

**325**   the height of 10km, while the red lines are closer to 0 than the black lines from the height of 10km

**326**   to 35km. It may be due to the non-difference method  cut down the noise level on the excess phase

**327**   paths and thereby obtains less-biased refractivity with in regions of multipath signal propagation

**328**   in the lower troposphere compared to single-difference technique. In addition, we also provide the

**329**   statistics results between R_N and "atmPrf" and ECMWF analyses in Table 5.

**330**

**331**   **Table 5.** The summary of the mean relative refractivity deviation between COSMIC observations and ECMWF

**332**                 analyses. (%)

| Parameter | status | ecmPrf | eraPrf | echPrf |
|---|---|---|---|---|



| | | | | |
|---|---|---|---|---|
| R_N | rising | 1.82 | 1.46 | 1.49 |
| | setting | 1.32 | 1.34 | 1.06 |
| atmPrf | rising | 1.61 | 1.20 | 1.19 |
| | setting | 0.99 | 1.04 | 0.62 |

Table 5 presents that the accuracy of R_N is slightly worse than "atmPrf" compared to ECMWF

analyses data. The mainly reason is subject to the effect of COSMIC receiver clock. Figure 8

shows the 5th and 6th COSMIC receiver clock error on December 12, 2013.

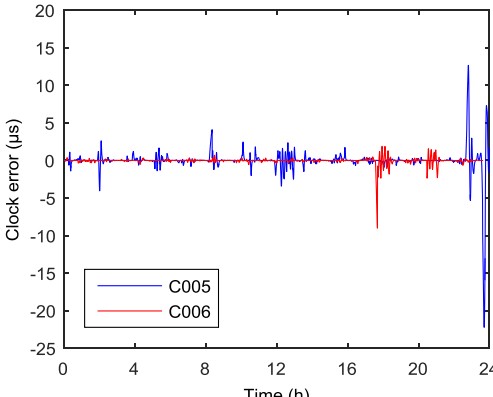

**Fig. 8.** The 5th and 6th COSMIC receiver clock error on December 12, 2013. "C005" represents the 5th COSMIC

RO satellite; "C006" represents the 6th COSMIC RO satellite.

Figure 8 shows that both the 5th and 6th COSMIC satellite receiver clock offsets are very

dramatic change, and it suggests that the COSMIC receiver without an ultra-stable oscillator.

Moreover, it can be seen that the maximum clock difference between the neighboring two epochs

is over 20 micro second from Figure 8. Therefore, it will bring in large noise when interpolating

the COSMIC clock offset solutions into the temporal resolution of 20ms. Then, these noises will

spread the excess Doppler and affect the accuracy of the refractive. In this study, the process of

AEP for each COSMIC occultation event utilizing non-difference technique will be discarded

when appearing a larger COSMIC clock oscillator with the same time of the RO event.

**5 Conclusions**

This study focused on the extraction of the AEP using the non-difference processing strategy.

Firstly, the COSMIC POD processing is used to accurately determine the position and velocity of

the center of mass of the satellite and the receiver offset based on PANDA software. Then,



according to the UCAR/CDAAC provided "opnGps" profiles, taking the gravitational delay error,
the relativistic effects, the receiver clock error and the phase center offsets into account, the
atmospheric excess phases can be estimated with the help of precise Final GPS orbits and
transmitter clock offsets from GPS time using non-difference approach. Finally, the bending angle,
refractive and dry temperature profiles are performed from AEP using ROPP software. Next, the
refractivity profiles obtained from non-difference method are validated by using "atmPrf" profiles.
The case analyses of representative rising and setting occultation events indicate that the relative
refractive offset between ND-derived and "atmPrf" profile are better than ±2% below the height of
30km while the relative refractive offset gradually increases with the altitude from the height of
30km to 40km. In addition, the average relative refractive deviation of globally distributed 200
events between ND-derived and "atmPrf" profiles show that the comparison results are changing
from ±0.5% to ±6% in setting RO events, and from ±0.5% to ±7% in rising RO events. The
statistical results of refractivity and dry temperature are better than 2.0% and 3.3%, respectively.
Finally, the mean relative refractivity deviation between COSMIC observations and ECMWF
analyses present that non-difference approach reduces the noise level on the excess phase paths in
the lower troposphere compared to single-difference processing strategy.  Subject to the impact of
receiver clock oscillation, the atmospheric excess phase process will fail using the non-difference
processing strategy in partial RO events. If the second-generation COSMIC receiver equipped an
ultra-stable oscillator, it would improve the quality of AEP using ND technique. Not only the
accuracy and the resolution of the results of the LEO POD, but also the accuracy of the refractive
can be improved in the future research.

**Referneces**
Ashby, N.: Relativity in the global positioning system, Living Rev. Relativity., 6, 1, 2003..

378        http://www.livinggreviews.org/lrr-2003-1.

Beyerle, G., Schmidt, T., Michalak, G.: GPS radio occultation with GRACE: Atmospheric

380        profiling utilizing the zero difference technique, Geophysical Research Letters., 32, L1386,

381        2005.



Bi, Y. M., Chen, J., Yang, G. L., Liao, M., Wu, R. H.: GPS occultation excess phase computed
utilizing the updated single difference technique, Acta Phys. Sin., 61(14), 149301, 2012.
CDAAC.: Algorithms for Inverting radio occultation signals in the neutral atmospheric research,
2005a. http://cosmic-io.cosmic.ucar.edu/caddc/doc/index.html.
CDAAC.: Algorithm description for LEO precision orbit determination with Bernese v5.0 at
CDAAC, COSMIC Project Office, University Corporation for Atmospheric Research., 2005b.
http://cosmic-io.cosmic.ucar.edu/caddc/doc/index.html.
COSMIC      Operations      Group.:      CDAAC      Data      Products,      2013.      http://cdaac-
www.cosmic.ucar.edu/cdaac/products.html.
Esteban, E., Vazquez, B., Borota, A., Grejner, B.: GPS-PWV estimation and validation with
radiosonde data and numerical weather prediction model in Antarctica, GPS Solut., 17, 29-

393      39, 2013.

Hajj G.A., Kursinski, E.R., Romans, L. J., Bertiger, W.I., Leroy, S.S.: A technical description of
atmospheric sounding by GPS occultation, J Atmos Sol Terr Phys.  64(4)., 451-469, 2002.
Hwang, C. W., Tseng, T. P., Lin, T. J.: Precise orbit determination for the FORMOSAT-
3/COSMIC satellite mission using GPS, Journal of Geodesy.,  83, 477-489, 2009.
Kuo, B., Rocken, C., Anthes, R.: GPS radio occultation missions, The second Formosat3/
COSMIC Data Users Workshop, Boulder, Colorado., 2007.
Kursinski, E. R., Hajj, G. A., Schofield, J. T.: Observing earth's atmosphere with radio
Occultation measurements using the Global Positioning System, J. Geophs. Res., 102(D19),

402      23429-23465, 1997.

Liu, J.N., Zhao, Q.L., Ge, M.R.: Preliminary result of CHAMP orbit determination with PANDA
software, The International Symposium on GPS/GNSS. Sydney Australia., 2004.
Montenbruck, O., Garcia-Fernandez, M., Williams, J. Performance comparison of semi-codeless
GPS receivers for LEO satellites, GPS Solut., 10, 249-261, 2006. Doi:10.1007/s10291-006-

407      0025-9

Rocken, C., Anthes, R., Exner, M.: Analysis and Validation of GPS/MET Data in the neutral
atmosphere,  Journal of Geophysical Research., 102(D25), 29849-29866, 1997.
Rocken, C., Kuo, Y. H., Schreiner, W.: COSMIC system description, special issue of terrestrial,



atmospheric and oceanic science., 11(1), 21-52, 2000.
Rocken, C., Anthes, R., Exner, M., Hunt, D., Sokolovskiy, S., Ware, R.: Analysis and validation of
GPS/MET data in the neutral atmosphere, J Geophys Res., 102(D25), 29849-29866, 1997.
Wilson, J., Anderson, C., Baker, M.: Radiometric calibration of the advanced wind scatter meter
radar ASCAT carried onboard the METOP-A satellite, IEEE Transactions on Geoscience and
Remote Sensing,  48(8), 3236-3255, 2010.
Ware, R.: GPS Sounding of the atmosphere from low earth orbit: Preliminary results, Bull. Am.
Meteorol. Soc., 77, 19-40, 1996.
Wickert, J., Beyerle, G., Hajj, G., Schwieger, V., Reigber, C.: GPS radio occultation with
CHAMP: atmospheric profiling utilizing the space-based single difference technique, Geophys.
Res. Lett., 29, 1187, 2002.
Wickert, J., Reigber, C., Beyerle, G., Konig, R.: Atmosphere sounding by GPS radio occultation:
first results from CHAMP, Geophysical Research Letters.,  28(17), 3263-3266, 2001.
Wickert, J.: GPS radio occultation with CHAMP and GRACE: A first look at a new and promising
satellite configuration for global atmospheric sounding, Ann. Geophys., 23, 653-658, 2005.
Wu, B. H., Fu, C. L., Liou, Y. A., Chen, W. J., Pan, H. P.: Quantitative analysis of the errors
associated with orbit uncertainty for FORMOSAT-3, In: Proceeding of the international
symposium on remote sensing (ISRS)., 87-90, 2005.
Schreiner, W., Rochen, C., Hunt, D.: Approach and quality assessment of single difference
processing of GPS radio occultation data at the UCAR CDAAC, A43A-0066, AGU Fall
Meeting, San Francisco, CA, December 11-15., 2005.
Schreiner, W., Rocken, C.: Quality assessment of COSMIC/FORMOSAT-3 GPS radio occultation
data derived from single-and double-difference atmospheric excess phase proccesion, GPS
Solut.,  14, 13-22, 2010.
Schreiner, B.: COSMIC GPS POD and limb antenna test report. Internal report of UCAR, 2005.
Schreiner, W., Rochen, C., Hunt, D.: Approach and quality assessment of single difference
processing of GPS radio occultation data at the UCAR CDAAC,  Eos Trans. AGU Fall
Meeting,  San Francisco, Abstract A43a-0066., 2005.
Schreiner, W., Sokolovskiy, S., Hunt, D., Rocken, C., Kuo, Y. H.: Analysis of GPS radio



occultation data from the FORMOSAT-3/COSMIC and Metop/GRAS missions at CDAAC,
Atmos. Meas.Tech., 4, 2255-2272, 2011.
Senior, K., Ray, J., Beard, R.: Characterization of periodic variations in the GPS satellite clocks,
GPS Solut., 12, 211-225., 2008. Doi:10.1007/s 10291-008-0089-9
Sokolovskiy, S., Hunt, D.: Statistical optimization approach for GPS/MET data inversions,
presentation at URSI GPS/MET Workshop, Tucson, AZ., 1996.
Yen, N. L., Huang. Chen, J. F.: FORMOSAT-3/COSMIC GPS radio occultation mission:
preliminary results, TEEE T. Geosci, Remote., 45, 3813-3825, 2007.
Yunck, T. P., Bertiger, W. I., Wu, S. C.: First assessment of GPS-based reduced dynamic orbit
determination on TOPEX/POSEIDON, Geophysical Research Letter., 21 (7), 541-544, 1994.