# Peer review of "Estimation and Evaluation of COSMIC Radio Occultation Excess Phase Using Non-differenced Measurements"

_Atmospheric Measurement Techniques, 2016_

## Short Comment (SC1) · 2 Dec 2016

Which method did you use in the ROPP software to process the radio occultation data of low troposphere?

How to interpolate the 1s clock error to a resolution of 0.02s?

---

## Author Comment (AC1) · 2 Dec 2016

In this study, a non-differenced (ND) processing strategy is proposed to estimate the AEP. To begin with, we used PANDA (Positioning and Navigation Data Analyst) software to perform the precise orbit determination (POD) for the COSIMC (The Constellation Observing System for Meteorology, Ionosphere and Climate) satellite to acquire the position and velocity of the center of mass of the satellite and the corresponded receive clock offset. The bending angles, refractivity and dry temperature profiles are derived from the estimated AEP by the ROPP (Radio Occultation Processing Package) software. The ND method is validated by the COSMIC products in typical rising and setting occultation events.In addition, we also compared the relative refractivity

bias between ND-derived and "atmPrf" profiles of globally distributed 200 COSMIC occultation events on December 12, 2013. The observed COSMIC refractivity profiles from ND processing strategy are further validated using European Centre for Medium-Range Weather Forecasts (ECMWF) analyses data, and the results indicate that non-differencing reduces the noise level on the excess phase paths in the lower troposphere compared to single difference processing strategy.

---

## Author Comment (AC2) · 2 Dec 2016

Reply to Biyan Chen:

Thank you very much for your comments, then i will reply your questions:

1ãĂĄComputation of bending angles below 25km by wave optics in the ROPP software.

2ãĂĄWe used Lagrange interpolation method to interpolate the 1 s clock error into 20 ms.

Thank you very much again!

---

## Referee Comment (RC1) · Anonymous Referee #1 · 24 Jan 2017

The manuscript presents non-differenced technique to calculate atmospheric excess phase (AEP) and validates the calculated AEP using data available from UCAR/CDAAC. Overall, the manuscript has high scientific significance and quality, however, it needs a through grammatical correction before publication. A detailed review is provided as an attachment.

Please also note the supplement to this comment:
http://www.atmos-meas-tech-discuss.net/amt-2016-276/amt-2016-276-RC1-supplement.pdf

[Figure]

**Supplement:**

Estimation and Evaluation of COSMIC Radio Occultation Excess Phase Using Non-Differenced Measurements

The article titled 'Estimation and evaluation of COSMIC radio occultation excess phase using non-differenced method' presents the non-differenced technique to calculate atmospheric excess phase (AEP), and compare the refractivities using these AEP to those obtained from UCAR/CDAAC, which uses single differenced method. Overall, the article has scientific merit and needs minor grammatical corrections for clarity. It is suitable for publication after minor corrections.

General comments and suggestions:

- The references need to be reorganized. They are not listed in alphabetical or chronological order, and in-text citation of the reference is ambiguous in some cases.
- In the 'Results validation and analysis' section, the authors compare collocated measurements and retrievals with UCAR/CDAAC. Does the collocated imply that the same pair of GNSS transmitters and LEO receivers are used? If the same transmitter-receiver pair is used, then it would be easier to the reader if this information is mentioned in the text instead of just stating collocated measurements.
- A comparison of the AEP from ND technique with 'atmPhs' would be interesting because it looks like refractivity difference between Ref_ND and Ref_Phs has a positive bias of ~0.5 % in setting occultation case (Fig 1) and a negative bias of ~ -0.5% in the rising occultation case (Fig 2). Is there a similar bias in the AEP from the ND method and 'atmPhs'?
- The average differences (Table 4) show >1.5% difference between refractivities derived using AEP from ND technique and AEP from 'atmPhs' using the ROPP software for the retrieval. In the troposphere, the difference is ~±0.5% in both the rising and setting cases (Fig 4 and Fig 5). What can be the contributing factor for this difference? Just by comparing the figures, one of the factors seem to be the excess phase difference. However, the difference between the ROPP and UCAR retrievals for the same excess phase also have differences of 0.51 and 0.93 % for setting and rising occultations, respectively, indicating the role of factors other than the excess phase.

Specific comments:

P6L148 – acronym 'COD' is not defined in the text.

P6L157 – 'experience force' is this typographical error?

P6L169 – Replace '3$^{th}$' with '3$^{rd}$'.

P13L295 – 'ecmPrf' is repeated. One of them should be changed to 'echPrf'.

P18L384 – Reference is not used in the text.

P18L386 – Reference is not used in the text.

P19L412 – This reference is already listed in P18L408, with less co-authors.

P20L442 – This reference does not appear in the text.

P20L448 – This reference does not appear in the text.

---

## Short Comment (SC2) · 13 Feb 2017

Dear Reviewer,

First of all, we would like to thank the anonymous reviewer very much. All the comments helped us improve the manuscript a lot. We are very appreciative of that. For each comment, we have carefully examined and answered with our best efforts. The paper is significantly revised and structured based on the reviewer's valuable comments and suggestions. Thank you! Please kindly find enclosed our updated manuscript and our responses to each comment below.

General comments and suggestions:

 c The references need to be reorganized. They are not listed in alphabetical or chronological order, and in-text citation of the reference is ambiguous in some cases. Response: Thank you very much! Revision has been made. The references are reorganized and listed in alphabetical order. Moreover, the in-text citation of references which are ambiguous or not used in the text are corrected.

 c In the 'Results validation and analysis' section, the authors compare collocated measurements and retrievals with UCAR/CDAAC. Does the collocated imply that the same pair of GNSS transmitters and LEO receivers are used? If the same transmitter-receiver pair is used, then it would be easier to the reader if this information is mentioned in the text instead of just stating collocated measurements. Response: Yes, the collocated imply that we used the same pair of GNSS transmitters and LEO receivers. We had added this information in the text. "We used the same transmitter-receiver pair to compare the collocated measurements and retrievals with UCAR/CDAAS."

 c A comparison of the AEP from ND technique with 'atmPhs' would be interesting because it looks like refractivity difference between Ref_ND and Ref_Phs has a positive bias of ∼0.5 % in setting occultation case (Fig 1) and a negative bias of ∼ -0.5% in the rising occultation case (Fig 2). Is there a similar bias in the AEP from the ND method and 'atmPhs'? Response: It is very difficult to obtain the absolute value of AEP using ND method or SN method. Because of the AEP usually contains ambiguity and time-independent error terms. However, we are interested in deriving the atmospheric Doppler (the time derivative of the atmospheric excess phase) from AEP. And the atmospheric Doppler obtained from the ND method and "atmPhs" has a similar bias with the refractivity derived from Ref_ND and Ref_Phs.

 c The average differences (Table 4) show >1.5% difference between refractivities derived using AEP from ND technique and AEP from 'atmPhs' using the ROPP software for the retrieval. In the troposphere, the difference is ∼±0.5% in both the rising and setting cases (Fig 4 and Fig 5). What can be the contributing factor for this difference? Just by comparing the figures, one of the factors seem to be the excess phase

difference. However, the difference between the ROPP and UCAR retrievals for the same excess phase also have differences of 0.51 and 0.93 % for setting and rising occultations, respectively, indicating the role of factors other than the excess phase. Response: The main sources of error above 30km are the incomplete ionospheric correction and the receiver tracking error, and the error below 10km (in the troposphere) is mainly due to the fact that atmospheric water vapor ambiguity can't be determined. The ROPP software and CDAAC software used different methods to deal with these these problems.

Specific comments: P6L148 – acronym 'COD' is not defined in the text. Response: We defined the acronym 'COD' (Center for Orbit Determination in Europe).

P6L157 – 'experience force' is this typographical error? Response: Thank you very much! Revision has been made. 'experience force' is modified to 'empirical acceleration'.

P6L169 – Replace '3th' with '3rd'. Response: Revision has been made.

P13L295 – 'ecmPrf' is repeated. One of them should be changed to 'echPrf'. Response: Thank you very much! Revision has been made.

P18L384 – Reference is not used in the text. Response: Thank you very much! Revision has been made.

P18L386 – Reference is not used in the text. Response: Thank you very much! Revision has been made.

P19L412 – This reference is already listed in P18L408, with less co-authors. Response: Revision has been made. Rocken, C., Anthes, R., Exner, M., Hunt, D., Sokolovskiy, S., Ware, R., Gorbunov, M., Schreiner, W., Feng, D., Herman, B., Kuo, Y.-H., Zou, X.: Analysis and Validation of GPS/MET Data in the neutral atmosphere, Journal of Geophysical Research., 102(D25), 29849-29866, 1997.

P20L442 – This reference does not appear in the text. Response: Thank you very

much! Revision has been made.

P20L448 –This reference does not appear in the text. Response: Thank you very much! Revision has been made.

Please also note the supplement to this comment:
http://www.atmos-meas-tech-discuss.net/amt-2016-276/amt-2016-276-SC2-supplement.pdf

---

## Referee Comment (RC2) · Anonymous Referee #2 · 27 Feb 2017

The manuscript compares a non-differenced technique for estimating atmospheric excess phase (AEP) then compares the results with the AEP provided in current GPS-RO processing system. Thogh, the topic is relevant to AMT, the manuscript needs a major revision for both English as well as the section that discusses the results.

1. Please revise the abstract and especially properly define ROPP and atmProf.

2. P3-L72, L75, L76 - please define and provide a reference

3. P5 - L136 - Bernes 5.2?

4. P10 L236; please use actual definitions rather than the color of the lines to explain the results. Please do so for the rest of the manuscript.

5. The manuscript currently only mentions the differences between different techniques but doesn't provide any error analysis. Please provide as much discussion as possible on the results and for instance explain why the differences are very large at higher altitudes. Please also compare your results with previous studies when possible.

6. Please only include the references that are actually cited.

7. Figures need a revision as the thickness of the lines is so small that hardly show up in the printed versions.

8. Please also consider revising the text for English.